# Etiology of Clinical Community-Acquired Pneumonia in Swedish Children Aged 1–59 Months with High Pneumococcal Vaccine Coverage—The TREND Study

**DOI:** 10.3390/vaccines9040384

**Published:** 2021-04-14

**Authors:** Annika Eklundh, Samuel Rhedin, Malin Ryd-Rinder, Maria Andersson, Jesper Gantelius, Giulia Gaudenzi, Magnus Lindh, Ville Peltola, Matti Waris, Pontus Nauclér, Andreas Mårtensson, Tobias Alfvén

**Affiliations:** 1Pediatric Emergency Department, Sachs’ Children and Youth Hospital, S-118 83 Stockholm, Sweden; samuel.rhedin@ki.se (S.R.); tobias.alfven@ki.se (T.A.); 2Department of Global Public Health, Karolinska Institutet, S-171 77 Stockholm, Sweden; giulia.gaudenzi@ki.se; 3Department of Medical Epidemiology and Biostatistics, Karolinska Institutet, S-171 77 Stockholm, Sweden; 4Pediatric Emergency Department, Astrid Lindgren Children’s Hospital, Karolinska University Hospital, S-171 64 Solna, Sweden; malin.ryd-rinder@sll.se; 5Department of Women’s and Children’s Health, Karolinska Institutet, S-171 77 Stockholm, Sweden; 6Department of Infectious Diseases, University of Gothenburg, S-405 30 Gothenburg, Sweden; maria.andersson.3@gu.se (M.A.); magnus.lindh@microbio.gu.se (M.L.); 7Department of Protein Science, Division of Nanobiotechnology, KTH Royal Institute of Technology, SciLifeLab, S-171 65 Solna, Sweden; jesper.gantelius@gmail.com; 8Department of Paediatrics and Adolescent Medicine, Turku University Hospital, University of Turku, Fi-20521 Turku, Finland; vilpel@utu.fi; 9Institute of Biomedicine, University of Turku and Clinical Microbiology, Turku University Hospital, Fi-20521 Turku, Finland; mwaris@utu.fi; 10Division of Infectious Diseases, Department of Medicine, S-17176 Stockholm, Sweden; pontus.naucler@ki.se; 11Karolinska Institutet & Department of Infectious Diseases, Karolinska University Hospital, S-17164 Solna, Sweden; 12Department of Women’s and Children’s Health, International Maternal and Child Health (IMCH), Uppsala University, S-75237 Uppsala, Sweden; andreas.martensson@kbh.uu.se

**Keywords:** pneumococcal conjugate vaccines, bacterial pneumonia, viral pneumonia, respiratory infection, etiology, children, World Health Organization

## Abstract

(1) Immunization with pneumococcal conjugate vaccines has decreased the burden of community-acquired pneumonia (CAP) in children and likely led to a shift in CAP etiology. (2) The Trial of Respiratory infections in children for ENhanced Diagnostics (TREND) enrolled children 1-59 months with clinical CAP according to the World Health Organization (WHO) criteria at Sachs’ Children and Youth Hospital, Stockholm, Sweden. Children with rhonchi and indrawing underwent “bronchodilator challenge”. C-reactive protein and nasopharyngeal PCR detecting 20 respiratory pathogens, were collected from all children. Etiology was defined according to an a priori defined algorithm based on microbiological, biochemical, and radiological findings. (3) Of 327 enrolled children, 107 (32%) required hospitalization; 91 (28%) received antibiotic treatment; 77 (24%) had a chest X-ray performed; and 60 (18%) responded to bronchodilator challenge. 243 (74%) episodes were classified as viral, 11 (3%) as mixed viral-bacterial, five (2%) as bacterial, two (0.6%) as atypical bacterial and 66 (20%) as undetermined etiology. After exclusion of children responding to bronchodilator challenge, the proportion of bacterial and mixed viral-bacterial etiology was 1% and 4%, respectively. (4) The novel TREND etiology algorithm classified the majority of clinical CAP episodes as of viral etiology, whereas bacterial etiology was uncommon. Defining CAP in children <5 years is challenging, and the WHO definition of clinical CAP is not suitable for use in children immunized with pneumococcal conjugate vaccines.

## 1. Introduction

Community-acquired pneumonia (CAP) is a major cause of mortality in children, but the burden has decreased following the introduction of immunization with pneumococcal conjugate vaccine (PCV) [1,2] that has now been implemented in most countries in the world. This has also led to a shift in CAP etiology, with a relative increase of viral CAP [3,4,5,6].

Defining etiology in childhood CAP is complex due to limitations in microbiological diagnostic methods and difficulties in obtaining representative specimens from the lower respiratory tract; moreover, in the majority of episodes of suspected bacterial CAP, antimicrobial treatment is prescribed without an identified causative agent [7,8]. Current prevention and treatment strategies are largely based on findings from pneumonia etiology studies performed during the 1970s and 1980s, where *Streptococcus pneumoniae* and *Haemophilus influenzae* were identified as the two major pathogens driving infection [9,10]. Recent CAP etiology studies in the PCV era, which have mostly been performed in low- and middle-income countries, have indicated that respiratory viruses, predominantly respiratory syncytial virus (RSV), influenza virus, and human metapneumovirus, account for the majority of CAP episodes in children and have reported low rates of atypical bacterial etiology [3,4,5]. Studies from high-income countries have presented contradicting data on etiology, partly due to differences in the classification of etiology [6,11,12]. There is also increasing evidence of complex interactions between the bacterial microbiome, the host immune system, and the infecting pathogen, suggesting that a large proportion of CAP episodes are in fact mixed viral-bacterial infections [13,14]. In Sweden, PCV immunization was implemented in the national child immunization program during 2008/2009 reaching high coverage (>97%) [15].

In 1991, the World Health Organization (WHO) published an algorithm with criteria for diagnosing CAP clinically. The purpose was to aid health personnel diagnose and treat CAP in resource-limited settings, and children fulfilling the criteria are recommended antimicrobial treatment with amoxicillin. The WHO criteria have most likely contributed to a decrease in the mortality of CAP but have also been criticized for having low specificity [16]. Despite this, the WHO criteria have been used clinically in other settings and also in several studies on pediatric pneumonia, in various socio-economic settings [4,9,10,17].

New studies on CAP etiology in varied settings are needed to improve our understanding of CAP in children as well as to guide empirical antimicrobial treatment of the disease [7]. The main objective of this prospective observational study was to study the etiology of clinical CAP in children below five years in a high-income country setting with high pneumococcal immunization coverage.

## 2. Materials and Methods

### 2.1. Study Design and Study Setting

The study protocol of the TREND study (Trial of Respiratory infections in children for Enhanced Diagnostics) has previously been described [18]. To summarize, the TREND study was a prospective observational study of children with clinical CAP. The study took place 19 November 2017–19 December 2019, at the emergency department of Sachs’ Children and Youth Hospital, Stockholm, a tertiary care level hospital with one of the largest pediatric emergency departments in Sweden, having over 30,000 visits each year. Children 1–59 months of age visiting the emergency department were eligible to participate in the study if they presented with symptoms consistent with clinical CAP according to WHO criteria: (1) breathing troubles or cough, and (2) observed age-adjusted tachypnea (≥50 breaths/min in children 1–11 months, ≥40/min in children ≥ 1 year) or chest indrawing. To increase the specificity of the WHO criteria, all cases with wheezing and chest indrawings underwent “bronchodilator challenge”, i.e., received inhalation therapy with a rapid acting bronchodilator, as suggested by the Pneumonia Etiology Research for Child Health study team (PERCH) [9]. For the inhalation, salbutamol (5 mg/mL) diluted with saline (9 mg/mL) was used and administered through a nebulizer: 2.5 mg Salbutamol for children ≤20 kg, 5 mg Salbutamol for children >20 kg. After inhalation, the cases were re-evaluated, and the “bronchodilator challenge” was considered positive if the chest indrawing had resolved [9]. Exclusion criteria were previous inclusion in the study and hospitalization during the last 14 days. The study was registered at clinicaltrials.gov (Last Update Posted: 23 November 2020) (ID:NCT03233516), 28 July 2017.

### 2.2. Sampling and Microbiological Analyses

Nasopharyngeal aspirates were collected from all cases within 24 h upon arrival at the emergency department. The nasopharyngeal aspirates were analyzed with real-time PCR at Clinical Microbiology, Sahlgrenska University Hospital, Gothenburg, using previously described methods [19]. The PCR detected the following 15 respiratory viruses and 5 bacteria: adenovirus, bocavirus, coronavirus (HKU1, NL63, OC43, 229E), enterovirus, Influenza A/B, RSV A/B, metapneumovirus, parainfluenza virus 1–3, rhinovirus, *Bordetella pertussis*, *Clamydophila pneumoniae*, *H. influenzae*, *Mycoplasma pneumoniae* and *S. pneumoniae*. Due to cross-reactivity between picornaviruses in the in-house real-time PCR, samples doubly positive for rhinovirus and enterovirus were coded as only positive for the pathogen with the lowest CT value if delta CT value > 5 cycles and else according to the clinical presentation as previously described [20]. Samples doubly positive for coronavirus OC43 and coronavirus HKU1 were coded as only positive for coronavirus OC43 if delta CT < 5 cycles. Capillary blood for rapid C-reactive protein (CRP) determination was collected at the emergency room and analyzed using the Alere Afinion™ AS100 Analyzer commercial kit. If multiple CRP tests were performed, the highest value < 48 h from arrival at the emergency unit was recorded.

### 2.3. Study Variables

All cases were examined according to a standardized case report form and the caregivers filled out a standardized electronic questionnaire with detailed information regarding the study subjects on, e.g., days of illness, current symptoms, number of siblings, vaccinations, antibiotic treatment, medication, chronic diseases, allergies, heredity for asthma, breastfeeding, smoking in the family, preschool, origin of parents, and socioeconomic status. Severe hypoxia was defined as a peripheral oxygen saturation <90%. Data on treatment, hospital, and intensive care admissions, as well as microbiological, radiological, and biochemical analyses were collected from the patient charts.

All X-ray images were reviewed by a senior consultant in radiology and classified according to the following criteria: normal examination/no infiltrates, other infiltrate, alveolar consolidation, pleural effusion, large dense infiltrate/lobar consolidation, evidence of empyema. Radiographic pneumonia as suggested by WHO was defined by the presence of alveolar consolidation and/or pleural effusion [21]. 

Etiology was classified according to the a priori defined TREND etiology algorithm [18], which was based on microbiological and biochemical findings (Figure 1). In short, bacterial etiology was defined as a positive blood/pleura culture, radiographic evidence of bacterial etiology (large/dense infiltrate or evidence of empyema) or high CRP (CRP > 80 mg/L in children <1 year or >120 mg/L in children 1–4 years) and the absence of codetection of influenza A/B, RSV, PIV, metapneumovirus or adenovirus by PCR. Viral and atypical bacterial etiology was defined according to nasopharyngeal PCR results and CRP. Less pathogenic viruses that frequently are detected in asymptomatic children were only considered as plausible etiologic agents if CRP was low (<20 mg/L) [22]. Bacterial PCR findings from the nasopharyngeal aspirates were not considered for the establishment of bacterial etiology. Mixed viral-bacterial etiology was defined as fulfilling the criteria for both viral and bacterial etiology. Children not fulfilling the criteria for any category was classified as having undetermined etiology.

### 2.4. Statistical Methods

For comparative analyses, the children were classified into three groups according to etiology: viral, bacterial/mixed/atypical, and undetermined, as the bacterial groups were too small to assess separately. 

Categorical data were analyzed using a chi-squared test or Fisher’s exact test where appropriate. Continuous data were assessed using a Kruskal-Wallis test or ANOVA, depending on the nature of the data. The following sensitivity analyses were performed with different diagnostic criteria: With or without exclusion of children responding to the bronchodilator challenge; diagnosis based on ICD-10 code (J10.0, J11.0 or J12–J18) by the treating physician; radiographic pneumonia according to WHO criteria (alveolar consolidation or pleural effusion). Two additional sensitivity analyses were performed that varied the etiology algorithm. First, the CRP cutoff for bacterial etiology was lowered to 60mg/L, in line with the PERFORM study [23]. Finally, the strict TREND algorithm that only considered microbiologically confirmed diagnosis was applied on the full cohort. Data were analyzed by using Stata version 16.1 (StataCorp, College Station, TX, USA) and R.

### 2.5. Ethics

The study was conducted in accordance with Good Clinical Practice and the latest version of the Declaration of Helsinki. Written informed consent was collected from the guardian(s) before inclusion in the study. The study was approved by the Regional Ethical Review Board in Stockholm (ref 2017/958-31). 

## 3. Results

### 3.1. Sociodemographic Characteristics 

After exclusion of 3 children with incomplete sampling, 327 children with clinical CAP according to WHO criteria were included in the study. The median age was 13 months (interquartile range 5 to 22), and 47% were 11 months or younger (Table 1). In total, 200 (61%) were male, and 107 (34%) had a chronic disease, predominantly asthma/wheezing, 91 (29%). Vaccination coverage was 99%, and 76% of the children had at least one parent with education corresponding to the university level. Children with viral etiology were younger and less likely to attend daycare, as compared to the other groups (*p* < 0.01 and *p* = 0.06, respectively) (Table 1).

### 3.2. Symptoms and Signs

In total, 291 (89%) of the children presented with tachypnea, 272 (83%) had chest indrawing, and 25 (8%) had severe hypoxia (peripheral oxygen saturation <90%). (Table 2). Only 138 (42%) presented with fever at the emergency department, but 247 (80%) children had a history of fever. Rhonchi/wheezing were present in 173 (54%), and nasal flaring was observed in 31 (10%) children. A total of 152 (46%) children presented with both chest indrawing and wheezing, of whom 139 (91%) underwent the “bronchodilator challenge” with a rapid acting bronchodilator, according to the study protocol. In 60 (43%) of these children, indrawing was resolved upon re-evaluation. Children with bacterial/mixed/atypical etiology were more likely to present with fever and severe hypoxia and less likely to present with indrawing, as compared to the other groups (Table 1). The median CRP value was 13 mg/L; 38 (12%) had a CRP value >60 mg/L, and 28 (9%) a CRP value >80 mg/L. Of the five children with bacterial etiology according to the TREND etiology algorithm, all presented with fever, four (80%) presented with tachypnea, three (60%) with indrawing and two (40%) had auscultatory rhonchi. None presented with nasal flaring or severe hypoxia. Three (60%) had an underlying chronic disease (two children had asthma/recurrent wheezing and one child had a chronic heart disease). 

### 3.3. Management and Treatment

Almost a third of the (*n* = 107, 32%) children required hospitalization, and 2 (0.6%) required transfer to the pediatric intensive care unit (PICU). No deaths occurred. Both children transferred to PICU tested positive for RSV; one of the children was also positive for *H. influenzae* and coronaviruses HKU1 and OC43. A chest X-ray was performed in 77 (24%) children; of these, 29 (38%) showed radiological evidence of CAP (alveolar consolidation or pleural effusion), but only 6 (8%) showed radiological evidence of lobar pneumonia. Antibiotics were started in 91 (28%) of the children, and 283 (87%) received inhalation treatment with either bronchodilator, saline or corticosteroids. Treatment with oxygen, high-flow nasal cannula and CPAP were started in 48 (15%), 28 (9%) and 4 (1%) children. Children with bacterial/mixed/atypical etiology were more likely to be treated with antibiotics (83% versus 25%, and 24% for viral and undetermined etiology, respectively, *p* < 0.001). They were also more likely to receive intravenous fluid treatment and to be admitted to inpatient wards, as compared to the other groups (*p* < 0.001 and *p* = 0.07, respectively) (see Table 2). Four (80%) of the five children with bacterial etiology received treatment with antibiotics. One child (20%) required hospitalization and none received treatment with oxygen, high-flow nasal cannula or CPAP.

### 3.4. Microbiological Findings 

A total of 310 children (95%) tested positive for at least one virus by PCR of nasopharyngeal swabs; the corresponding number for bacteria was 242 (74%). The most common viral PCR findings were rhinovirus (156 (48%)) and RSV (126 (39%)); the most common bacterial PCR findings were *S. pneumoniae* (179 (55%)) and *H. influenzae* (143 (44%)) (Figure 2). RSV was more commonly detected in children < 2 years (43% vs. 20% in children 2–4 years, *p* < 0.001) (Figure 2). *M. pneumoniae* was only detected in two children; one child tested positive for *B. pertussis* and none for *C. pneumoniae*. 

In 94 children (29%), more than one virus was detected, and co-detections of both viruses and bacteria by PCR were seen in 233 children (71%). Among the co-infections, the most common pathogen pairs were rhinovirus + *S. pneumoniae* (83 (25%)), RSV + *S. pneumoniae* (79 (24%)) and RSV + *H. influenzae* (66 (20%)) (Appendix A). A total of 15 blood cultures were collected; none were positive. A total of 13 children had radiographic signs of pleural effusion, but no one had empyema. No pleural sampling was performed. 

### 3.5. Etiological Classification

According to the TREND etiology algorithm, 243 (74%) episodes were classified as viral, 11 (3%) as mixed viral-bacterial, five (2%) as bacterial, two (0.6%) as atypical bacterial, and 66 (20%) as undetermined etiology (Figure 1). Of the five episodes classified as bacterial etiology, one had a chest X-ray showing lobar pneumonia, whereas the other four episodes were classified as bacterial etiology based on the high CRP. Bacterial PCR findings were not considered for the classification of etiology. However, *H. influenzae* was more commonly detected in children with mixed viral-bacterial etiology (detected in 73%, *p* = 0.035) and these children also had significantly lower CT-values, as compared to the other groups (mean CT = 25.8 versus 29.9 and 32.2 for viral and undetermined etiology respectively, *p* < 0.01). In contrast, there was no significant difference in the presence of or mean CT-values of *S. pneumoniae* between the different etiology groups.

### 3.6. Sensitivity Analyses

Children responding to the bronchodilator challenge (*n* = 60) were analyzed separately. These children were less likely to have mixed viral-bacterial etiology (0% vs. 4%) as compared to the rest of the children (Figure 3C). They were also older, more likely to have an underlying diagnosis of asthma or obstructive disease, and to test positive for rhinovirus as compared to the rest of the children (Appendix A). However, excluding these children had a non-significant effect on the etiology estimate as compared to the main analysis (Figure 3D). Further, when the analyses were restricted to children with the attending physicians’ CAP diagnoses, this resulted in an increased share of bacterial (8%) and mixed viral-bacterial (14%) etiology (Figure 3B). The same tendency was seen when restricting the analyses to children with radiographic pneumonia (Figure 3E). When the CRP cutoff for defining bacterial etiology was lowered to 60 mg/L, the shares of bacterial (3%) and mixed viral-bacterial (9%) etiology slightly increased, whereas the share of viral etiology decreased (69%) (See Appendix A). Finally, the strict TREND algorithm was applied, which classified 179 (55%) episodes as viral and 148 (45%) as undetermined etiology (Appendix A).

## 4. Discussion

In this prospective observational study of children aged 1–59 months with high PCV coverage presenting with clinical CAP according to the WHO criteria at a tertiary care level pediatric emergency unit in Stockholm, Sweden, we report that viral etiology accounted for the large majority of included episodes, whereas bacterial etiology was uncommon and atypical bacterial etiology almost non-existent, when applying the novel TREND etiology algorithm. The small number of children with bacterial etiology may reflect the effect of a high PCV vaccination coverage. This has implications for the empirical treatment and management of children with clinical CAP in this setting. Our findings are in line with other CAP etiology studies performed in high-income countries with high PCV coverage. Jain et al. reported viral etiology in 66% of US children with CAP and bacterial or mixed viral-bacterial etiology in 8% and 7%, respectively [6]. Berg et al., a Norwegian study of children <18 years with radiographic pneumonia, reported viral etiology in 63% and pneumococcal etiology in 11% [12]. Our lower estimate of bacterial etiology is likely explained by differences in case definition, as well as the fact that the other two studies included children up to the age of 18 years, whereas we only included children below five years. By using a pragmatic case definition based on clinical presentation, we might have enrolled an unknown number of children with other lower respiratory tract infections that did not have true CAP. Studies of CAP etiology from low- and middle-income countries have had conflicting results, partly owing to differences in definition of bacterial etiology [4,5,10]. We defined etiology according to an a priori defined etiology algorithm, based on microbiological, biochemical, and radiological findings, which classified <10% of children with clinical CAP as bacterial etiology [18]. We performed several different sensitivity analyses and observed that the proportion of bacterial etiology was higher when restricting the analyses to children with a discharge diagnosis of pneumonia or radiographic evidence of CAP, but still bacterial etiology accounted for less than a third of all episodes. 

Of note, *S. pneumoniae* was detected in the respiratory tract by PCR in >50% and *H. influenzae* in >40% of the children. The clinical significance of upper respiratory tract testing of bacteria among children aged 1–59 months is limited, due to a large degree of asymptomatic infection, and we did not consider bacterial upper respiratory testing when assigning etiology [6,10]. However, previous studies have reported interaction between *S. pneumoniae* and *H. influenzae* with potentially increased severity. [24,25] When assessing the 83 (25%) children who tested positive for both *S. pneumoniae* and *H. influenzae* separately, we found evidence of higher CRP compared to the rest of the children (mean CRP 35 mg/L versus 23 mg/L, respectively, *p* < 0.01) but no increased risk of hospitalization. PCV vaccine studies and studies of the respiratory microbiota have suggested that viral-bacterial interactions are important for disease severity and likely mixed viral-bacterial etiology of CAP is underestimated [13,26]. Indeed, we found that children with mixed viral-bacterial etiology had an increased likelihood to test positive for *H. influenzae* and also to have lower CT-values indicating a higher bacterial load. We also observed clusters of RSV/*H.influenzae* and of rhinovirus/*S. pneumoniae* in line with previous studies [13,27]. More studies on viral-bacterial interactions are needed to improve our understanding of pediatric CAP. Atypical bacteria were rarely detected by PCR in our study, suggesting that routine antimicrobial coverage of atypical agents in pediatric non-severe CAP in children aged 1–59 months in our setting and similar ones should be discouraged. 

Almost a third of the children had an asthma diagnosis underscoring the risk of misclassification bias in studies of clinical CAP, as there is an overlap in clinical presentation between obstructive disease and CAP. To account for this, “bronchodilator challenge” was performed on children presenting with rhonchi and wheezing as suggested by the PERCH study team [9]. Children responding to bronchodilator challenge were overrepresented in terms of underlying asthma diagnosis, and a large majority were classified as having viral or undetermined etiology, suggesting that the bronchodilator challenge could be considered for use in clinical praxis to improve the specificity of CAP diagnostics. However, excluding these children did not have a significant impact on the etiology classification.

Interestingly, less than a third of the children received antibiotic treatment and only 15% were diagnosed by the treating physician as CAP. A large number of children with viral-induced wheeze and/or viral bronchitis were also included using the WHO algorithm despite the addition of the bronchodilator challenge. This underscores that the WHO algorithm for clinical CAP, in our high-income setting, appears inappropriate for children immunized with PCV and would result in inappropriate antibiotic treatment [18,28,29,30]. In Sweden, the use of antibiotics is fairly restrictive and regulated, yet >20% of children with clinical CAP classified as viral etiology received antibiotic treatment in the study, indicating that there is still room for improvement for antimicrobial stewardship. Further, almost 20% of children classified as bacterial/mixed viral-bacterial or atypical bacterial etiology, according to the TREND etiology algorithm recovered without antibiotic treatment. Although some of these children might have had a viral infection, it is also important to keep in mind that some bacterial CAP episodes are in fact self-limiting, and randomized controlled trials in low- and middle-income countries have shown limited benefit from amoxicillin treatment for non-severe clinical CAP [29,31]. 

A strength of the study was the prospective design and standardized data collection with an a priori defined algorithm of etiology classification. A limitation was the fact that we did not routinely perform blood cultures or chest X-rays. Hence, we could have misclassified some children with bacterial CAP as viral or undetermined etiology if they had a low CRP. Yet, we believe this number to be small, as pediatric blood cultures have poor sensitivity and chest X-rays are routinely performed on children with severe disease [12]. Further, not every child meeting the inclusion criteria was included in the study. Due to logistical reasons (mainly the handling of the microbiological samples), no enrollment took place during the night or on weekends/holidays. Another limitation was the lack of a proper control group, as some viruses are frequently detected in asymptomatic children, which complicates the interpretation of viral detections [22]. It is possible that some of the episodes were misclassified as viral or mixed viral-bacterial etiology due to incidental detection of less pathogenic viruses not related to the episode. The WHO criteria were originally developed for use in low-income settings (by non-medical professionals) and have previously been criticized for having low specificity, but also low sensitivity, since the respiratory rate is such an important inclusion criterion. This may have resulted in a number of missed cases with bacterial pneumonia, since they do not always present with an increased respiratory rate [17]. Lastly, we did not have data on pneumococcal serotypes and hence cannot distinguish between PCV and non-PCV serotypes in the children, yet previous studies on pneumococcal carriage in Sweden have indicated that PCV types are rare in children [32].

The study underscores the complexity of defining CAP in children, and also the difficulty in finding the causative agent solely by standard diagnostic methods as there is no perfect reference standard for bacterial CAP. To improve the proper antibiotic use, there is a need for novel biomarkers.

## 5. Conclusions

The novel TREND etiology algorithm classified the majority of episodes of clinical CAP in children aged 1–59 months as being of viral etiology, whereas bacterial and atypical bacterial etiologies were uncommon. Defining CAP in children <5 years is complex, and the WHO definition of clinical CAP appears inappropriate for use in children immunized with PCV in high-resource settings, as it would result in inappropriate antibiotic prescription.

## Figures and Tables

**Figure 1 vaccines-09-00384-f001:**
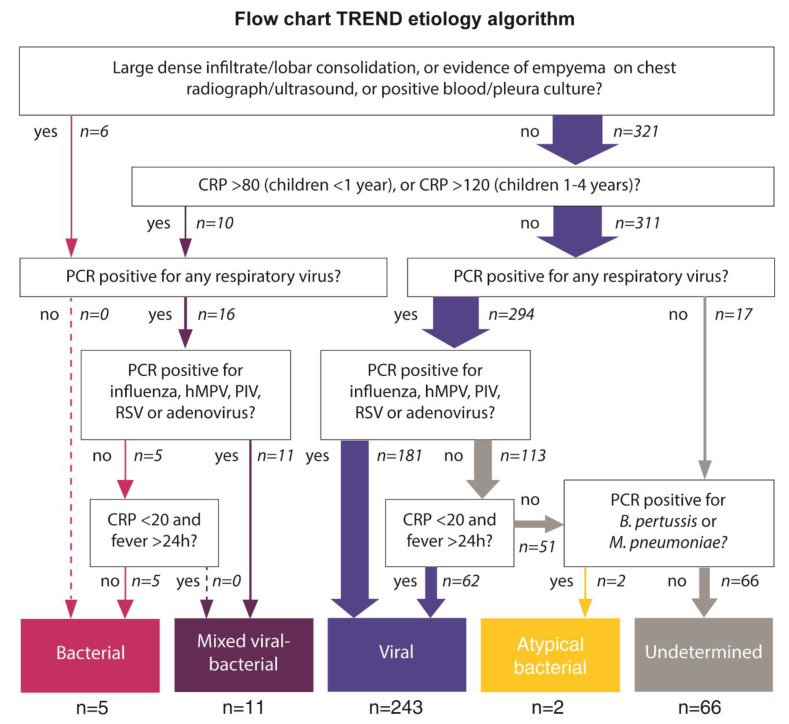
Flowchart of the TREND etiology algorithm. Abbreviations: *B. pertussis*, *Bordetella pertussis*; CRP, C-reactive protein; h, hour; hMPV, human metapneumovirus; *M. pneumoniae*, *Mycoplasma pneumoniae*; PCR, polymerase chain reaction; PIV, parainfluenza virus; RSV, respiratory syncytial virus; TREND, Trial of Respiratory infections for Enhanced Diagnostics.

**Figure 2 vaccines-09-00384-f002:**
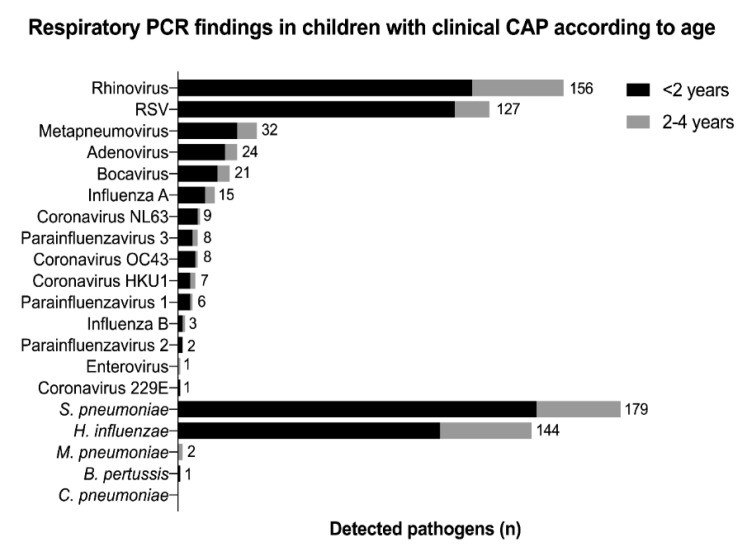
Real-time PCR findings according to age in nasopharyngeal aspirates of children with clinical CAP in the TREND study (*n* = 327). Numbers showing the total number of detected pathogens. Abbreviations: *B. pertussis*, *Bordetella pertussis*; *C. pneumoniae*, *Chlamydophila pneumoniae*; CAP, community-acquired pneumonia; *H. influenzae*, *Haemophilus influenzae*; *M. pneumoniae*, *Mycoplasma pneumoniae*; PCR, polymerase chain reaction; RSV, respiratory syncytial virus; *S. pneumoniae*, *Streptococcus pneumoniae*; TREND, Trial of Respiratory infections for Enhanced Diagnostics.

**Figure 3 vaccines-09-00384-f003:**
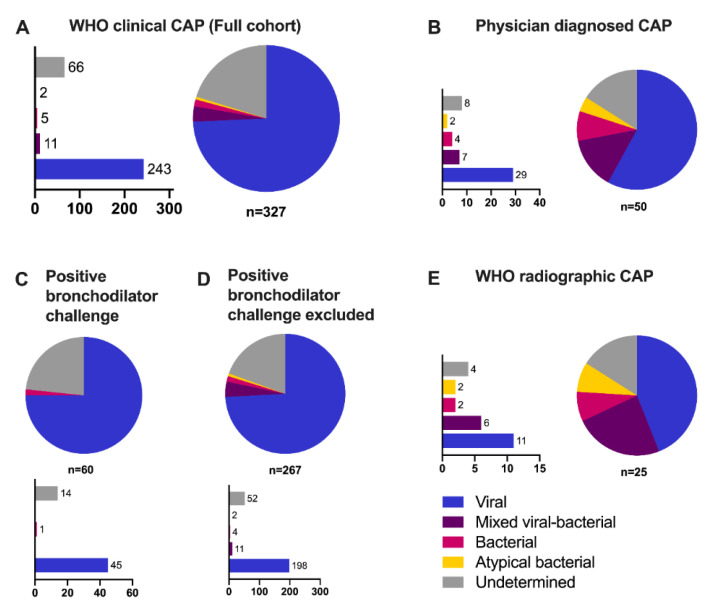
Etiology classification of children with clinical CAP in the TREND study in (**A**) the full cohort based on WHO criteria (*n* = 327), (**B**) children with physician-diagnosed CAP (ICD-10 code of J10.0, J11.0 or J12-J18) (*n* = 50), children (**C**) with (*n* = 60) or (**D**) without (*n* = 267) positive bronchodilator challenge and (**E**) children with WHO radiographic CAP (*n* = 25). Abbreviations: CAP, community-acquired pneumonia; WHO, World Health Organization.

**Table 1 vaccines-09-00384-t001:** Sociodemographic characteristics and clinical presentation of study participants according to the TREND etiology algorithm classification.

Characteristic	Viral Etiology (*n* = 243)	Bacterial/ Mixed Viral-Bacterial/ Atypical Etiology (*n* = 18)	Undetermined (*n* = 66)	All (*n* = 327)	*p*-Value
**Characteristics**					
Age (months), median IQR	11 (4–18)	17 (8–26)	20 (9–26)	13 (5–22)	<0.01
*1–11 months*	*126 (52)*	*7 (39)*	*20 (30)*	*153 (47)*	
*12–59 months*	*117 (48)*	*11 (61)*	*46 (70)*	*174 (53)*	
Male sex	149 (61)	8 (44)	43 (65)	200 (61)	
Attending daycare	92 (38)	10 (59)	33 (52)	135 (42)	0.058
Breastfeeding	83 (35)	5 (28)	18 (28)	106 (33)	0.503
Parental smoking	36 (15)	1 (6)	8 (12)	45 (14)	0.496
Fully immunized ^a^	238 (99)	17 (94)	65 (100)	320 (99)	0.23
Chronic disease	73 (31)	6 (35)	28 (45)	107 (34)	0.236
*Asthma/wheezing*	*61 (26)*	*5 (29)*	*25 (40)*	*91 (29)*	
*Other* ^b^	*12 (5)*	*1 (6)*	*3 (5)*	*16 (5)*	
University studies ≥1 parent	176 (72)	16 (89)	55 (83)	247 (76)	0.075
**Clinical presentation**					
Tachypnea ^c^	213 (88)	16 (89)	62 (94)	291 (89)	0.394
Chest indrawings	204 (84)	12 (67)	56 (85)	272 (83)	0.182
Positive bronchodilator challenge ^d^	45 (19)	1 (6)	14 (21)	60 (18)	0.312
Cough	236 (98)	17 (100)	60 (97)	313 (98)	0.752
Breathing troubles	166 (79)	11 (69)	45 (83)	222 (79)	0.441
Fever, ≥38° C	103 (42)	14 (78)	21 (32)	138 (42)	0.002
History of fever	184 (80)	18 (100)	45 (78)	247 (80)	0.066
Rhonchi/wheezing	132 (55)	3 (17)	38 (58)	173 (54)	0.005
Peripheral oxygen saturation <90%	11 (5)	6 (33)	8 (13)	25 (8)	<0.001
Nasal flaring	23 (10)	1 (6)	7 (11)	31 (10)	0.802
Grunting	14 (6)	2 (11)	6 (9)	22 (7)	0.512
CRP mg/L median (IQR)	9 (0–24)	124,5 (97–165)	22 (10–40)	13 (0–33)	*NA*
*CRP <20 mg/L*	67 (28)	16 (89)	41 (62)	*124 (38)*	*NA*
*CRP >60mg/L*	18 (7)	14 (78)	6 (9)	*38 (12)*	*NA*
*CRP >80mg/L*	10 (4)	14 (78)	4 (6)	*28 (9)*	*NA*

Numbers expressed as n (%) if not otherwise specified. ^a^ According to age. ^b^ Congenital syndrome (e.g., Trisomi 21); *n* = 2, chronic heart/lung disease, *n* = 5; neurological disease/epilepsy, *n* = 2; other disesase (not specified), *n* = 8, ^c^ Age-adjusted (≥50 breaths/min in children 1–11 months, ≥40 breaths/min in children ≥1 year). ^d^ Performed in 139/152 (91%) of children with indrawings and auscultatory rhonchi. Abbreviations: CRP, C-reactive protein; IQR, interquartile range; NA, Not Applicable.

**Table 2 vaccines-09-00384-t002:** Management of study participants according to the TREND etiology algorithm classification.

Management/Treatment	Viral Etiology (*n* = 243)	Bacterial/ Mixed Viral-Bacterial/ Atypical Etiology (*n* = 18)	Undetermined (*n* = 66)	All (*n* = 327)	*p*-Value ^a^
Antibiotics	60 (25)	15 (83)	16 (24)	91 (28)	<0.001
Oral corticosteroids	60 (25)	2 (12)	24 (36)	86 (26)	0.06
Inhalation treatment	213 (88)	12 (67)	58 (88)	283 (87)	0.039
Intravenous fluid	14 (6)	7 (39)	8 (12)	29 (9)	<0.001
Nasogastric tube	20 (8)	3 (17)	4 (6)	27 (8)	0.317
Oxygen treatment	30 (12)	5 (28)	13 (20)	48 (15)	0.091
High-flow nasal cannula	21 (9)	2 (11)	5 (8)	28 (9)	0.872
CPAP	4 (2)	-	-	4 (1)	0.667
Chest X-ray	45 (19)	15 (83)	17 (26)	77 (24)	<0.001
*Normal*	*5 (11)*	*1 (7)*	*4 (24)*	*10 (13)*	
*Other infiltrate*	*30 (67)*	*5 (33)*	*10 (59)*	*45 (58)*	
*Alveolar consolidation*	*10 (22)*	*3(20)*	*3 (18)*	*16 (21)*	
*Large dense infiltrate/* *lobar consolidation*	*-*	*6 (40)*	*-*	*6 (8)*	
*Pleural effusion*	*4 (9)*	*6 (40)*	*3 (18)*	*13 (17)*	
*Evidence of empyema*	*-*	*-*	*-*	*-*	
Admitted to hospital ward	71 (29)	8 (44)	28 (42)	107 (32)	0.071
Admitted to ICU	1 (0,4)	1 (6)	-	2 (0.6)	0.147

^a^ Comparing the three different etiology groups. Abbreviations: CI, confidence interval, CRP, C-reactive protein, CPAP, continuous positive airway pressure.

## Data Availability

The data presented in this study are available on request from the corresponding author.

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
