# Peer review of "Etiology of Clinical Community-Acquired Pneumonia in Swedish Children Aged 1–59 Months with High Pneumococcal Vaccine Coverage—The TREND Study"

_vaccines, 2021, doi:10.3390/vaccines9040384_

Round 1
Reviewer 1 Report
In this article, the authors deal about the Etiology of Clinical Community-Acquired Pneumonia in Swe- 2 dish Children Aged 1-59 months with High Pneumococcal Vaccine Coverage.
The manuscript is well written, but I have a few comments for improving it:
-In the Introduction section, data about more germs including more severe bacterias which could determine CAP are missing. The authors mentioned only Streptococcus pneumoniae and Haemophilus influenza. I suggest adding it. (doi.org/10.3892/ETM.2018.6737, doi.org/10.3892/MMR.2016.6034, doi.org/10.3892/MMR.2017.7746). Aslo.
- More data about the ethical framework of testing vaccines (doi: 10.1007/s40199-020-00371-8) and the safety of vaccines should be added (doi.org/10.1016/j.toxrep.2020.10.016)
-What are the clinical pitfalls of this study?
Consider revising accordingly!
Author Response
Response to reviewer comments - Etiology of clinical community-acquired pneumonia in Swedish children aged 1-59 months with high pneumococcal vaccine coverage – the TREND study
We would like to thank the reviewer for valuable comments and believe that the manuscript has now been improved. Please see below our responses to the reviewer comments.
Reviewer 1
Comment: In this article, the authors deal about the Etiology of Clinical Community-Acquired Pneumonia in Swedish Children Aged 1-59 months with High Pneumococcal Vaccine Coverage.
The manuscript is well written, but I have a few comments for improving it:
Response: We are very happy to hear that reviewer 1 finds the study to be well written.
Comment: In the Introduction section, data about more germs including more severe bacterias which could determine CAP are missing. The authors mentioned only Streptococcus pneumoniae and Haemophilus influenza. I suggest adding it. (doi.org/10.3892/ETM.2018.6737, doi.org/10.3892/MMR.2016.6034, doi.org/10.3892/MMR.2017.7746). Aslo.
Response: We agree with reviewer 1 that other bacteria than S. pneumoniae and H. influenzae (such as Mycoplasma pneumoniae and Staphylococcus aureus) can cause CAP in children. Nevertheless, previous etiological studies on pediatric CAP have indicated that S. pneumoniae still accounts for the majority of bacterial CAP episodes and that atypical etiology is uncommon in children <5 years (O’Brien et al Lancet 2019, Jain et al NEJM 2015).
We have now added a sentence about atypical etiology in the introduction as follows:
P2, R54-61: “Current prevention and treatment strategies are largely based on findings from pneumonia etiology studies performed during the 1970s and 1980s, where Streptococcus pneumoniae and Haemophilus influenzae were identified as the two major pathogens driving infection [9,10]. Recent CAP etiology studies in the PCV era, which have mostly been performed in low- and middle-income countries, have reported low rates of atypical etiology and have indicated that respiratory viruses, predominantly respiratory syncytial virus (RSV), influenza virus and human metapneumovirus, account for the majority of CAP episodes in children [3-5].
Comment: More data about the ethical framework of testing vaccines (doi: 10.1007/s40199-020-00371-8) and the safety of vaccines should be added (doi.org/10.1016/j.toxrep.2020.10.016)
Response: While we agree with reviewer 1 that the ethical framework of testing vaccines and vaccine safety are truly important questions to further improve child health we believe that it is slightly beyond the scope of this article.
Comment: What are the clinical pitfalls of this study?
Response: There are several clinical pitfalls when studying CAP etiology.
We have previously discussed some of the limitations of the study as follows:
P11, R335-351: “A limitation was the fact that we did not routinely perform blood cultures or chest x-rays. Hence, we could have misclassified some children with bacterial CAP as viral or undetermined etiology if they had a low CRP. Yet, we believe this number to be small, as pediatric blood cultures have poor sensitivity and chest x-rays are routinely performed on children with severe disease [12]. Further, not every child meeting inclusion criteria was included in the study. Due to logistical reasons (mainly the handling of the microbiological samples), no enrollment took place during the night or on weekends/holidays. Another limitation was the lack of a proper control group, as some viruses are frequently detected in asymptomatic children, which complicates the interpretation of viral detections [22]. It is possible that some of the episodes were misclassified as viral etiology due to incidental detection of less pathogenic viruses not related to the episode. The WHO criteria were originally developed for use in low-income settings (by non-medical professionals) and have previously been criticized for having low specificity, but also low sensitivity, since the respiratory rate is such an important inclusion criterion. This may have resulted in a number of missed cases with bacterial pneumonia, since they do not always present with an increased respiratory rate [17].
We have now added a sentence on the limitation of the pragmatic case definition:
P10, R283-286: “Our lower estimate of bacterial etiology is likely explained by differences in case definition, and by the fact that the other two studies included children up to the age of 18 years, whereas we only included children below five years. By using a pragmatic case definition based on clinical presentation, we likely included a large number children with other lower respiratory tract infections that did not have true CAP.
Reviewer 2 Report
- I wondered if the authors had any data on the serotypes of the 179 pneumococcal samples (ln 221), or perhaps this has been published or will be published as a separate study elsewhere? The impact of PCV on pneumococcal carriage is well-established; data on this 179 will inform on circulating serotypes (presumably non-PCV) within the population.
- It appears that concurrent carriage of Sp and Hi (a common occurrence: Lewnard et al., 2016) can be a predictor of pneumonia (Chochua et al., 2016). What proportion of this study's cohort were positive for both Sp and Hi (presumably, concurrent carriage)? And I wondered if you could comment on whether the Chochua et al. data was unique to that cohort, and if not, is there any benefit in adding concurrent carriage of Sp and Hi to the etiology algorithm?

Author Response
Response to reviewer comments - Etiology of clinical community-acquired pneumonia in Swedish children aged 1-59 months with high pneumococcal vaccine coverage – the TREND study
We would like to thank the reviewer for valuable comments and believe that the manuscript has now been improved. Please see below our responses to the reviewer comments.
Reviewer 2:
Comment: I wondered if the authors had any data on the serotypes of the 179 pneumococcal samples (ln 221), or perhaps this has been published or will be published as a separate study elsewhere? The impact of PCV on pneumococcal carriage is well-established; data on this 179 will inform on circulating serotypes (presumably non-PCV) within the population.
Response: We agree with reviewer 2 that data on specific pneumococcal serotype would have been of great interest to better understand the beneficial effect of PCV in this setting. Previous studies on pneumococcal carriage in Sweden have indicated that the pneumococcal carriage in children is largely unchanged due to replacement of vaccine types with non-vaccine types (Lindstrand et al Vaccines 2016). Unfortunately we did not have data on pneumococcal serotypes in this study as the bacterial upper respiratory testing was PCR-based. We have now discussed this as a limitation of the study as follows:
P 12, R 351: “Lastly, we did not have data on pneumococcal serotypes and hence cannot distinguish between PCV and non-PCV serotypes in the children, yet previous studies on pneumococcal carriage in Sweden have indicated that PCV types are rare in children (29).”
Comment: It appears that concurrent carriage of Sp and Hi (a common occurrence: Lewnard et al., 2016) can be a predictor of pneumonia (Chochua et al., 2016). What proportion of this study's cohort were positive for both Sp and Hi (presumably, concurrent carriage)? And I wondered if you could comment on whether the Chochua et al. data was unique to that cohort, and if not, is there any benefit in adding concurrent carriage of Sp and Hi to the etiology algorithm?
Response: We thank reviewer 2 for this comment and the interesting studies by Lewnard et al and Chochua et al. We have now added the suggested references and assessed children doubly positive for HI and SP (n=83, 25%). There is some evidence of a more severe disease (higher CRP) in these children. We believe, however, that there is a need for more studies supporting this before implementing it in the etiology algorithm. We have now discussed this further as follows:
P11, R294-298: Of note, S. pneumoniae was detected in the respiratory tract by PCR in >50% and H. influenzae in >40% of the children. The clinical significance of upper respiratory tract testing of bacteria among children aged 1-59 months is limited, due to a large degree of asymptomatic infection, and we did not consider bacterial upper respiratory testing when assigning etiology [6,10]. However, previous studies have reported interaction between S. pneumoniae and H. influenzae with potentially increased severity [24, 25]. When assessing the 83 (25%) children who tested positive for both S. pneumoniae and H. influenzae separately we found evidence of higher CRP compared to the rest of the children (mean CRP 35mg/L versus 23mg/L respectively, p<0.01) but no increased risk for hospitalization. PCV vaccine studies and studies of…
Round 2
Reviewer 1 Report
No answer given.